# Student reactions to traumatic material in literature: Implications for trigger warnings

**Matthew Kimble**[1]*, **William Flack**[2], **Jennifer Koide**[1], **Kelly Bennion**[3], **Miranda Brenneman**[4], **Cynthia Meyersburg**[5]

1 Department of Psychology, Middlebury College, Middlebury, Vermont, United States of America,
2 Department of Psychology, Bucknell University, Lewisburg, Pennsylvania, United States of America,
3 Department of Psychology and Child Development, California Polytechnic State University, San Luis Obispo, California, United States of America, 4 Department of Psychology, Coastal Carolina University, Conway, South Carolina, United States of America, 5 Foundation for Individual Rights in Education (FIRE), Philadelphia, Pennsylvania, United States of America

* mkimble@middlebury.edu

**Data Availability Statement:** The data underlying this study contains sensitive participant information and cannot be shared publicly according to the original participant consent agreements. Further data inquiries should be

## Abstract

### Introduction

While trigger warnings have garnered significant debate, few studies have investigated how students typically respond to potentially triggering material.

### Method

In this study, three hundred and fifty-five undergraduate students from four universities read a passage describing incidences of both physical and sexual assault. Longitudinal measures of subjective distress, PTSD symptoms, and emotional reactivity were taken.

### Results

Greater than 96% of participants read the triggering passage even when given a non-triggering alternative to read. Of those who read the triggering passage, those with triggering traumas did not report more distress although those with higher PTSD scores did. Two weeks later, those with trigger traumas and/or PTSD did not report an increase in trauma symptoms as a result of reading the triggering passage.

### Conclusions

Students with relevant traumas do not avoid triggering material and the effects appear to be brief. Students with PTSD do not report an exacerbation of symptoms two weeks later as a function of reading the passage.

directed to the Middlebury College IRB
(IRB@middlebury.edu).

**Funding:** MK R15 MH081276 National Institute of
Mental Health https://www.nimh.nih.gov/index.
shtml CM John Templeton Foundation https://
www.templeton.org/ The funders had no role in
study design, data collection and analysis, decision
to publish, or preparation of the manuscript.

**Competing interests:** The authors have declared
that no competing interests exist.

## Introduction

In 2016, The University of Chicago publicly announced its opposition to trigger warnings in academic and social settings [1]. The University argued that the presence of trigger warnings allowed individuals to "retreat from ideas and perspectives at odds with their own [2]." This position contributed to ongoing discussion regarding the potential benefits and costs of trigger warnings, and questioned the psychological and intellectual consequences for college students [3–5].

In academia, trigger warnings are statements that warn students, particularly those with clinical mental health issues, about potentially disturbing material [1, 6]. They are common in academic settings and are frequently used to preface content that could be emotionally upsetting. The warnings may be fairly brief, for example, simply stating the type of experience referred to (i.e., "Trigger Warning: Domestic Violence"). Conversely, they can be extensive, alerting the student to potential negative reactions as well as mental health resources in the community. Most frequently, trigger warnings have been used to preface material related to significant traumas such as sexual assault, but they have also been extended to cover a wider range of issues, such as racism and misogyny, that individuals might find upsetting [4].

Ideally, trigger warnings allow individuals who have experienced trauma to be forewarned about material that may elicit unwanted memories of past traumatic events [7]. These memories may prompt strong physiological reactions and unpleasant emotions, responses with enough research support to be codified into the DSM-5 symptoms of posttraumatic stress disorder (PTSD) [8]. By forewarning students about triggering material, instructors allow individuals who have experienced trauma time to prepare or avoid altogether potential emotional distress. For example, a student may 1) choose to sit in the back of the class so they can easily slip out if they become upset, 2) read or watch material at a time or a place that is most advantageous for them or 3) set an appointment with a counselor to prepare or debrief about the material. Wyatt has argued that giving trigger warnings is an empathic act that helps minimize distress for students with trauma-related histories [9].

However, some have hypothesized that trigger warnings have more costs than benefits for students, even for those with relevant traumas [4, 5]. While a trigger warning may allow individuals to benefit from avoiding potentially distressing material, arguably they also preclude coping with their trauma, potentially adding to their psychological distress [7]. In addition, theory related to expectancy effects and self-fulfilling prophecies suggests that trigger warnings may make individuals more likely to respond poorly to upsetting material or lead them to interpret normal, brief, and appropriate distress as pathological [3, 5, 10]. In this scenario, trigger warnings would actually make students worse rather than better.

In addition, it appears that both instructor and student attitudes towards trigger warnings are mixed [11–14]. For example, when asked whether they supported the use of trigger warnings in their training, medical students were evenly split among the options [13]. Psychology students across both public and private institutions showed similar ambivalence, as some appreciated their potential value but were concerned that avoidance of triggering topics by instructors might limit their learning [13, 15]. In the end, the use of trigger warnings by instructors vary considerably and, short of decrees such as that by the University of Chicago, there is no consistent pattern across institutions or instructors regarding their use.

Research could inform this debate, but the research is only emerging. In the first of two studies, Bellet, Jones, and McNally assigned online participants to distressing passages with and without trigger warnings [3]. They found that participants who received the trigger warning believed themselves to be more vulnerable if they were to experience a trauma in the future than those who did not receive a trigger warning. Those individuals in the trigger warning

group also reported more anxiety related to the passage, but only if they endorsed the idea that words can cause harm. However, this effect was not replicated by the authors in a subsequent paper [16]. In fact, the authors argue that the most reliable finding was that trigger warnings did not work as intended, in that they may actually cause small increases in anxiety consistent with previous work [17]. In sum, this emerging body of work suggests some small and subtle effects regarding the impact of trigger warnings on emotional responses.

Kimble, Koide, and Flack (2017) reported on a sample of 123 undergraduates, including individuals with trauma histories, who received either a positive, a negative, or a neutral trigger warning before reading a passage that involved physical and sexual assault [18]. All participants reported higher levels of distress (Subjective Units of Distress Scale) [19] right after reading the passage as compared to 2 and 14 days later. This suggests that the effects were fairly brief and that they occurred across all participants regardless of what type of trigger warning was received. The primary significant effect was that those who met criteria for a provisional PTSD diagnosis reported higher distress at all three time points, more likely a reflection of their ongoing PTSD symptoms than a function of their study participation. This finding was corroborated by an extensive and thorough report by Sanson and colleagues [20]. In six separate studies that looked at student reactions to readings and film clips, the authors found that participants responded similarly to the material regardless of whether trigger warnings were presented or not. The authors conclude that trigger warnings have little effect, either positively or negatively.

These studies leave many questions unanswered including the central question inherent to the use of trigger warnings: Does "triggering" material have a negative and lasting effect on individuals with a trauma and PTSD? Do such readings in fact "trigger" these individuals, cause significant distress, and make them think more about their own trauma than they otherwise would? Research has shown that trauma-related stimuli can reliably produce physiological and psychological effects in trauma survivors, but these studies typically use materials that are very evocative and/or idiographic [21]. In the research literature on PTSD, such priming paradigms have not included material that are typically at issue in academic circles.

The answer to whether or not one should give trigger warnings should be based at least in part on how individuals typically respond to such material. While most prior research has investigated the effects of this type of material and trigger warnings on an average student, there has yet to be a study that investigates how students with PTSD symptoms and relevant traumas respond to potentially triggering material. To some extent, this is the population of most interest as they would mostly likely be the ones who would be triggered. In addition, with the exception of the study by Kimble and colleagues, there have been no data on what this looks like over a period of weeks [18]. Psychological effects that are lasting should be given more consideration than effects that are brief.

The term "trigger" has another inherent assumption: that individuals with a trauma related to the presented material will respond more strongly and have an exacerbation of symptoms that is greater than those who experienced a trauma that is not related to the passage. If individuals are indeed having their own traumas triggered by the material, then an individual who has experienced an assault should respond more strongly to a passage about an assault than some other type of traumatic event. However, some have argued that traumatized individuals may be more broadly responsive to threat and, while responding strongest to material related to their trauma, they may still be responsive to a wide range of negative material [22, 23]. This generalized effect could be even greater in those diagnosed with PTSD.

Finally, there have been no data on whether students avoid this material in academic settings, possibly through requests for different assignments or absences from class [4, 5]. There is some suggestion, typically in less formal venues such as blogs and posts, that students will

use trigger warnings to excuse themselves from academic responsibilities or retreat to "safe spaces." While there is ample evidence to suggest that many individuals with PTSD use cognitive and behavioral strategies to avoid reminders of their trauma, there are only sporadic and anecdotal reports that students do this in the context of their courses. Whether students use warnings with any frequency to avoid triggering material is entirely unknown.

In the current study, students from four different universities read a passage that contained a depiction of physical and sexual assault and were tracked in their emotional responses and symptoms over a two-week period. The design of the study allowed us to answer the following questions:

Q1: What percentage of students avoid potentially triggering material when they have the opportunity to do so?

Q2: Does trauma type affect the amount of distress experienced by the student after reading a triggering passage?

Q3: Do individuals with PTSD or a triggering trauma have an exacerbation of PTSD-related symptoms after doing the assignment?

## Method

This work was approved by the Institutional Review Boards at Middlebury College, Bucknell University, California Polytechnic State University San Luis Obispo, and Coastal Caroline University. In all cases, the consent form was in written format. All analyses were completed using SPSS 27 (IBM SPSS Statistics for Windows, Version 27).

### Participants

Three hundred and fifty-five undergraduate students at four different universities participated in the study. All were taking an Introductory Psychology class and received course credit for participation. Only basic demographics were collected on the participants given that the Introductory Psychology participant pool was quite small at two of the institutions and demographics beyond gender and race/ethnicity could be identifying to the investigators and their research assistants. Sixty-eight (68%) percent were female and thirty-two percent (32%) were male. Primary racial and ethnic identification were as follows: White 69%, Black 10%, Asian 6%, Multi-racial 8%, Hispanic 6%, and Other 2%. Provisional PTSD rates (PCL-5 scores greater than 33) were 12.3% (44/355) and the rate of endorsement of at least one of the 17 traumas having "Happened to me" on the LEC-5 was 87% (308/355). The participating institutions ranged from small private residential liberal arts schools (Middlebury College, enrollment of 2,500; Bucknell University 3,600) to mid-size public universities (California Polytechnic Institute, San Luis Obispo, enrollment of 20,000; Coastal Carolina University, enrollment of 9,000). Table 1 includes demographics at these institutions with respect to Gender, Race, SES, and selectivity.

### Materials

**Readings.**    All participants read a passage from Toni Morrison's novel *The Bluest Eye* (originally published in 1970) [24]. Hard copies from the First Vintage International Edition were used for the readings. Passages from pages 39–45 and 154–163 were chosen as the "Trigger Passage" as they included descriptions of sexual and physical assault. A passage of equal length from pages 110 to 126 was chosen as the neutral passage as it contained no depictions of assault.

**Table 1. Institutional characteristics**[*]**.**

| School | % Female | % BIPOC[#] | % Pell Grant | % Admitted | Enrollment |
|---|---|---|---|---|---|
| California Polytechnic State University | 48 | 43 | 14 | 28 | 22,013 |
| Coastal Carolina University | 55 | 32 | 37 | 69 | 10,641 |
| Bucknell University | 51 | 23 | 10 | 34 | 3,668 |
| Middlebury College | 54 | 35 | 17 | 15 | 2,611 |

[*]Retrieved March 9, 2021 www.collegesimply.com
[#]Black, Indigenous, People of Color

**Life events checklist for DSM-5 [25].** The LEC-5 is a 17-item self-report inventory that screens for potentially traumatic event exposure. The 17th item asks whether the participant has ever experienced an extraordinarily stressful event that was not addressed in the previous 16 items. The questionnaire measures whether a participant has experienced a traumatic event, witnessed an event, learned about it, was part of their job, are not sure, or not applicable.

The LEC-5 proves to be a moderately reliable measurement particularly with respect to items that were experienced by an individual (https://www.ptsd.va.gov/professional/assessment/te-measures/life_events_checklist.asp). Because of the difficult nature of self-report, especially in conjunction with traumatic experiences, the validity of the LEC-5 has not yet been fully determined [26].

The LEC-5 was administered in hard copy on Day 1 of the study to evaluate histories of potentially traumatic events (PTE). Those individuals who endorsed items 6–9 (items regarding physical or sexual assault) as having "happened to me" were considered to have a "Trigger Trauma". Participants who indicated that any other item from the LEC-5 had "happened to me" were assigned to the "Other Trauma" group. Those who did not report that any of the events had happened to them were placed in the "No Trauma" group.

**Subjective Units of Distress Scale (SUDS) [19].** The SUDS is a widely used scale that measures the amount of distress someone feels in the moment [19]. We used an eleven-point scale that ranged from 0–10 with higher scores associated with greater magnitudes of distress. Participants completed the SUDS on all three survey days in order to determine current feelings of distress. SUDS has high convergent (Spearman rho = .21, $p < .05$), discriminant (Spearman rho = .28, $p < .05$), predictive (Spearman rho = .51, $p < .001$) and concurrent (Spearman rho = .46, $p < .001$) validity [19].

**PTSD checklist for DSM-5 (PCL-5) [27].** The PCL-5 is a 20-item self-report survey that assesses the DSM-5's 20 symptoms for posttraumatic stress disorder. Items are scored from 0 (not at all) to 4 (extremely). Higher scores are associated with greater magnitudes of PTSD symptoms with scores ranging from 0–80. A score of 33 is considered adequate for provisionally diagnosing an individual with PTSD. A 10- to 20-point score change often signifies a clinically significant change (unpublished data: https://www.ptsd.va.gov/professional/assessment/adult-sr/ptsd-checklist.asp). In this study, "P-PTSD" will be used to designate those individuals above 33 on the PCL-5 and thus considered to have a provisional PTSD diagnosis.

Similar to the psychometric properties of the original PCL, the PCL-5 has high convergent and discriminant validity, with Cronbach's alphas ranging from .76 to .97 [28, 29]. Participants completed the PCL-5 during Day 1 and Day 14 and did so with respect to their most traumatic event.

**Alternate PTSD checklist (Alt.PCL).** For the purposes of this study, the PCL was adapted to assess any symptoms specifically related to the reading. This alternative PCL (Alt.PCL) was

also administered on Days 2 and 14. The instructions on Day 2 read, "Below is a list of problems that people sometimes have in response to an emotional passage. Please read each problem carefully and then circle one of the numbers to the right to indicate how much you have been bothered by the problem in the last two days." Only thirteen items were used: items 1–7 and 15–20. These items correspond to the re-experiencing, behavioral avoidance, and hyperarousal clusters and were judged to be capable of being linked with some specificity to the reading itself. For example, item #1 asked participants to report "Repeated, disturbing, and unwanted memories of the passage" and item #4 asked about "Feeling very upset when something reminded you of the passage."

**Reactions to Research Participation Questionnaire-Revised (RRPQ-R) [30].** The RRPQ-R is a 24-item questionnaire that measures participants' reactions to engaging in a research study. Only items 3, 4, 5, 10 and 16 were used in this study, as they represented the Emotional Reactions Factor of the survey measure plus a question on insight and consisted of questions such as "I was emotional during the research session" and "The research raised emotional issues for me that I hadn't expected." The questionnaire uses a 5-point Likert scale—from strongly disagree (1) to strongly agree (5). Therefore, total scores ranged from 0 to 20 with higher scores indicating stronger emotional reactions to participation. The internal reliability of the RRPQ-R has a Cronbach's alpha in the range of .82-.83, which is moderate [30]. The reliability (Cronbach's alpha) in this sample was .80.

## Design

The study was a 2 x 3 x 3 repeated measures design with "Passage" (Trigger v. Control) and "Trauma Type" (Trigger v. Other v. None) as between-subjects factors and "Time" (Day 1, Day 2, and Day 14) as the within-subjects factor. Some subsequent analyses that focused on the reactions of those specifically with provisional PTSD (i.e., PCL-5 > 33) replaced "Trauma Type" with "PTSD" (Yes v. No) in the above ANOVA. Cells testing"PTSD" and "Other Trauma" (n = 11) and "No Trauma" (n = 2) were too small for reliable analyses of a "Trauma Type x PTSD" interaction. In addition, other analyses sometimes used only two time points (Day 1 v. Day 14 or Day 2 v. Day 14), rather than three, resulting in only two levels of the "Time" factor.

The number of students who avoided the passage (Question 1) was simply recorded as the percentage of individuals assigned to the triggering passage who voluntarily chose the control passage. Question 2 (the impact of trauma type on response to the passage) was evaluated using the repeated measures ANOVA with Trauma type as the between-subjects variable and Time as the within-subjects variable. A Trigger type x Time interaction on the SUDS and RRPQ-R would indicate that past triggering traumas differentially impact on the emotional responses to the passage. Question 3 (the impact of trauma type on PCL scores) was also evaluated using the repeated measures ANOVA with Trauma type as the between-subjects variable and Time as the within-subjects variable. A Trigger type x Time interaction on PCL scores would indicate that past triggering traumas increase PCL scores from Day 0 to Day 14 in those who have had a relevant trauma. This analysis would be the best evidence for "triggering," i.e. a trigger-related stimulus prompting an increase in PTSD symptoms related to a personal trauma.

The majority of participants (n = 319) were randomly assigned to the "Trigger Passage" as opposed to the "Control Passage" condition (n = 36) because the primary goal of the study was to assess how individuals of differing trauma types responded to the triggering material (not to the neutral material). For random assignment to condition, one in every ten participants was assigned to the control condition using a random number generator. Given that only a

proportion of participants would have the trigger trauma or PTSD, a fairly large sample was needed in the trigger trauma condition in order to generate adequate power to test some of the study's primary hypotheses. However, a small subset was randomly assigned to the "Control Passage" condition (n = 36) as a manipulation check to assure that the triggering passage was in fact more distressing than the neutral passage.

## Procedure

Procedures differed to some extent across the four different universities. However, all locations had the following in common:

**Day 1.** All participants came to a lab or a classroom and completed and signed a consent form. A trained investigator then gave them the Day 1 envelope which included a background questionnaire, the LEC-5, the PCL-5, the reading (either control reading or trigger reading), then the SUDS, and the RRPQ-R. The session, including the reading, took approximately 40 minutes.

All participants, even those assigned the trigger warning passage, had the option to read the control passage. Right before the reading, they were instructed to consider the following: "If you prefer, you can do an alternate reading of identical length which is located in your packet. It is also from the novel "The Bluest Eye" by Toni Morrison. This alternative passage has no content which would generally be considered triggering. The packet for this is located in the back of this envelope."

Therefore, any participant could avoid the triggering passage if that was their preference and read the alternate passage instead. This option allowed us to assess the percentage of students who would voluntarily avoid a potentially triggering passage if they could. In comparison, those assigned to the control reading only had the option to read the control passage.

**Day 2.** Participants received an online link in an email that was delivered approximately 48 to 72 hours after their Day 1 participation. The email linked to a Qualtrics (Seattle, WA) survey that contained the SUDS and Alt.PCL. They were instructed to find a quiet place to complete the survey where they would have few if any distractions. They were asked to complete the survey within 24 hours. The survey typically took less than 5 minutes to complete.

**Day 14.** Participants again received an online link in an email two weeks after they began the study that went to a survey containing the third and final SUDS, a PCL-5 (asking them about their posttraumatic symptoms in the past two weeks related to their index trauma), and the Alt.PCL asking about any symptoms specific to the reading. At one site, participants returned to the lab to complete this online assessment. After completing the measures, they were taken to a debriefing page containing contact information for the investigators.

## Results

### Removal of outliers/attrition

In order to avoid using invalid data, cases were removed in instances in which there were very large discrepancies between Time 1 and Time 2 data points for any given dependent variable. Endorsement patterns suggested that some participants quickly clicked through the Time 2 online assessment with little regard for accuracy. This was typically reflected in very low Time 1 scores (i.e., PCL score of 0) and extremely high and discrepant scores at Time 2 (i.e., PCL score of 48). To remove such outliers, a difference score from Time 1 to Time 2 was calculated for each dependent variable and those cases in which the difference score was greater than two SD's were removed from further analyses. This resulted in a loss of 15 participants for the PCL data and 25 for the Alt.PCL data.

Fifty-eight participants who completed the Day 1 assessment did not complete the PCL on the Day 14 assessment. This represents a dropout rate of 16.3%.

There was no significant difference in PCL scores on Day 1 between those who dropped out and those who did not, $t(344)$ = -0.19, $p$ = .85. There was also no significant difference between dropouts in Day 1 RRPQ-R scores, $t(347)$ = -1.61, $p$ = .11. This suggests that those with more PTSD symptoms on Day 1 were not more likely to drop out of the study than those with lower scores. Additionally, dropout rate did not differ based on trauma type, $x^2(2)$ = 0.24, p = .89, suggesting that those with a trigger trauma were just as likely to continue in the study as those without a trigger trauma.

## Q1: What percentage of students avoid potentially triggering material when they have the opportunity to do so?

**Avoidance of triggering material.** Of the 319 individuals assigned to read the triggering passage, 305 (95.6%) did so. Fourteen participants elected to read the alternative passage. This included 126 out of 130 of those who had a trigger trauma (96.9%) and 41 out of 42 (97.6%) of those with Day 1 PCL-5 scores of 33 or above (and thus qualifying for a provisional PTSD (P-PTSD) diagnosis).

## Q2: Does trauma type affect the amount of distress experienced by students after reading a triggering passage?

**RRPQ-R: Condition, trauma, and P-PTSD effects.** Those participants who were assigned and read the triggering passage demonstrated a trend with slightly higher RRPQ-R scores than those assigned to the control passage, $t(331)$ = -1.89, $p$ = .06, Cohen's d = -.33. RRPQ-R scores did not differ between groups in response to the triggering passage (n = 298) based on whether they had a triggering trauma, other trauma, or no trauma, F(2,295) = 2.14, p = .12, eta squared = .014). Those with P-PTSD had significantly higher RRPQ-R scores (n = 40) to the triggering passage than those with PCL-5 scores below 33 (n = 252). Equal variances using the Levene's test were not met so the reported t values do not assume equal variances between groups: $t(76.52)$ = -6.38, $p <$ .001, Cohen's d = -.77.

**SUDS scores: Condition, trauma type, and P-PTSD effects.** SUDS scores over time were assessed with a repeated measures ANOVA with Condition as the between-subjects variable (Trigger v. Control) and Time (Day 1, Day 2, Day 14) as the within-subjects variable. Greenhouse-Geisser corrections were used in this mixed model ANOVA when Mauchly's Test of Sphericity were significant. The trigger condition had an n = 249 and the control condition had an n = 32. SUDS scores differed as a function of Time, $F(1.7, 465.5)$ = 46.50, $p <$ .001, partial eta squared = 0.25 and Condition, $F(1,279)$ = 23.02, $p <$. 001, partial eta squared = .076, with lower scores occurring later and higher SUDS scores in response to the triggering passage. However, these main effects were modified by a Condition x Time interaction, $F(1.67,465.5)$ = 9.96, $p <$ .001, partial eta squared = .034. The Condition effect differed at Day 1, with those in the triggering condition reporting significantly higher SUDS than those reading the control passage (p < .001), an effect that dissipated by Day 2 ($p$ = .074) and remained non-significant at Day 14 ($p$ = .059).

In response to the triggering passages (n = 248), SUDS scores did not differ as a function of Trauma Type nor was there a Trauma Type x Time interaction, $F(3.4,416)$ = 0.25, $p$ = .88, partial eta squared = .002. SUDS scores did differ as a function of P-PTSD, with those with scores above 33 on the PCL-5 reporting higher SUDS scores at all three time points, $F(1,242)$ = 42.89, $p <$ .001, partial eta squared .15. There was no P-PTSD x Time interaction, however, as both groups' scores decreased over time, $F(1.7,411)$ = 0.29, $p$ = .71, partial eta squared = .01 (Fig 1).

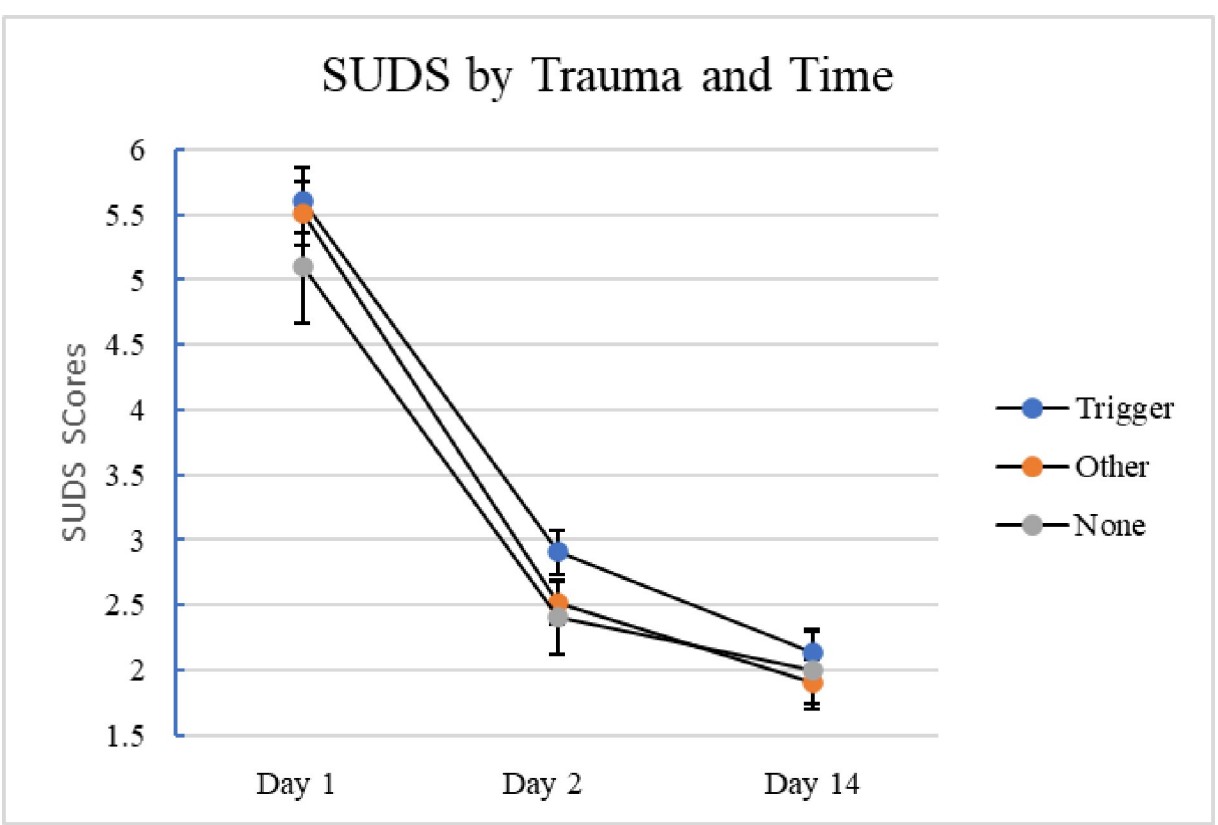

**Fig 1. SUDS by trauma and time.** Average subjective units distress shown with standard error bars as a function of day and trauma type (Trigger, n = 105; Other, n = 109, None, n = 34).

**Alternate PCL scores: Condition, trauma type, and provisional PTSD effects.** To test condition effects, alternate PCL scores over time were submitted to a repeated measures ANOVA with Condition as the between-subjects variable (Trigger v. Control) and Time (Day 2 v. Day 14) as the within-subjects variable. Analyses were based on an n = 250 after non completers and outliers were removed. There was a statistical trend for Condition, with individuals assigned to the Trigger Condition reporting more symptoms related to the reading across time but not significantly so, $F(1,248) = 3.45$, $p = .064$, partial eta squared = .014. Analysis by symptoms cluster (Reexperiencing, Avoidance, Hyperarousal) demonstrated that this trend was driven by a statistically significant effect for the hyperarousal cluster, in which those assigned to the trigger condition reported more hyperarousal symptoms related to the reading than those assigned to the control condition, $F(1,250) = 4.51$, $p = .035$, partial eta squared = .018. In no analyses was there a main effect for Time, nor was there a Condition x Time interaction. Using Trauma Type as a between-subjects variable in ALT.PCL scores to the triggering passage (n-217) revealed that those with a trigger trauma reported higher Alt.PCL scores regardless of Time, $F(2,214) = 3.35$, $p = .037$, partial eta squared = .03 (see Fig 2). Tukey's post hoc tests showed that the trigger trauma group had trended toward higher alternate PCL scores compared to the other trauma group ($p = .051$) and but were not signficantly different from the no trauma group ($p = .145$). Further analyses of the clusters indicated this pattern was present for primarily due to the hyperarousal cluster. There was no case in which the other trauma and no trauma groups differed from each other. There was no overall main effect for Time nor was

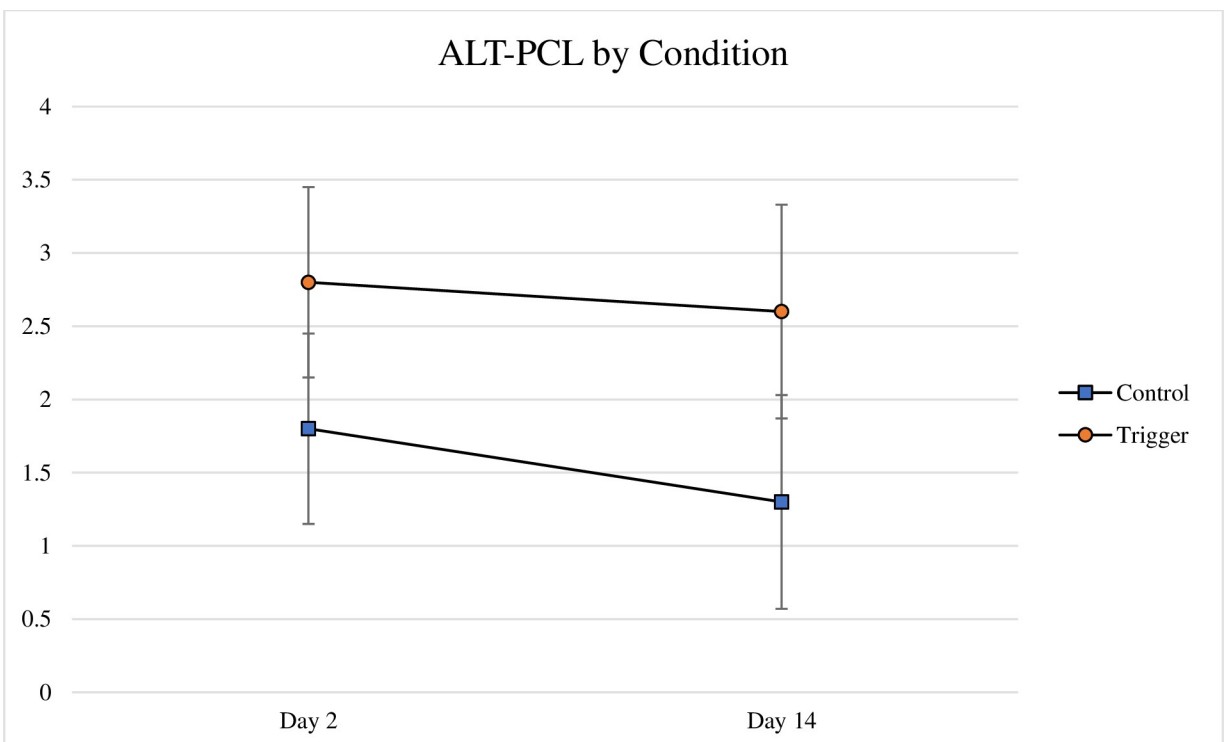

**Fig 2. Alt.PCL by trauma and time.** Average alternative PTSD checklist scores shown with standard error bars as a function of the control (n = 32) and triggering conditions (n = 218).

there a Trauma Type x Time interaction. To the triggering passage, using Provisional PTSD (n = 213) as a between-subjects factor showed a main effect for P-PTSD, $F(1,211) = 56.94$, $p < .001$, partial eta squared = .013, but not a P-PTSD x Time interaction, $F(1,211) = 2.82$, $p = .095$, partial eta squared = .013. There was no main effect for Time.

### Q3: Do individuals with PTSD or a triggering trauma have an exacerbation of PTSD related symptoms after doing the assignment?

**PCL-5 scores over time: Trauma type and PTSD effects.** PCL scores over time were submitted to a repeated measures ANOVA with Trauma Type as the between-subjects variable and Time (Day 1 and Day 14) as the within-subjects variable. Only individuals who received the triggering passage and who reliably completed the PCL at both time points were included in this analysis (n = 236). There was a main effect for Trauma Type, with those with a triggering trauma reporting higher PCL scores at Time 1 and Time 2 compared to both of the other groups, $F(2,233) = 10.93$, $p < .001$, eta squared = .086. Tukey's post hoc tests indicated that the other trauma group did not differ significantly from the no trauma group, $p = .59$. There was also a main effect for Time, with scores for all three groups lower at Time 2 than Time 1, $F(1,233) = 52.14$, $p < .001$, partial eta squared = .183. There were no significant interactions with time, $F(2,233) = 1.59$, $p = .21$, partial eta squared = .13, indicating that all three groups had lower scores on the PCL at Time 2 than Time 1. (Fig 3). P-PTSD participants did not differ over time in their reactions to the triggering passage either; there was no significant P-PTSD x Time interaction, $F(1,234) = 3.09$, $p = .080$, partial eta squared = .013. (Fig 4).

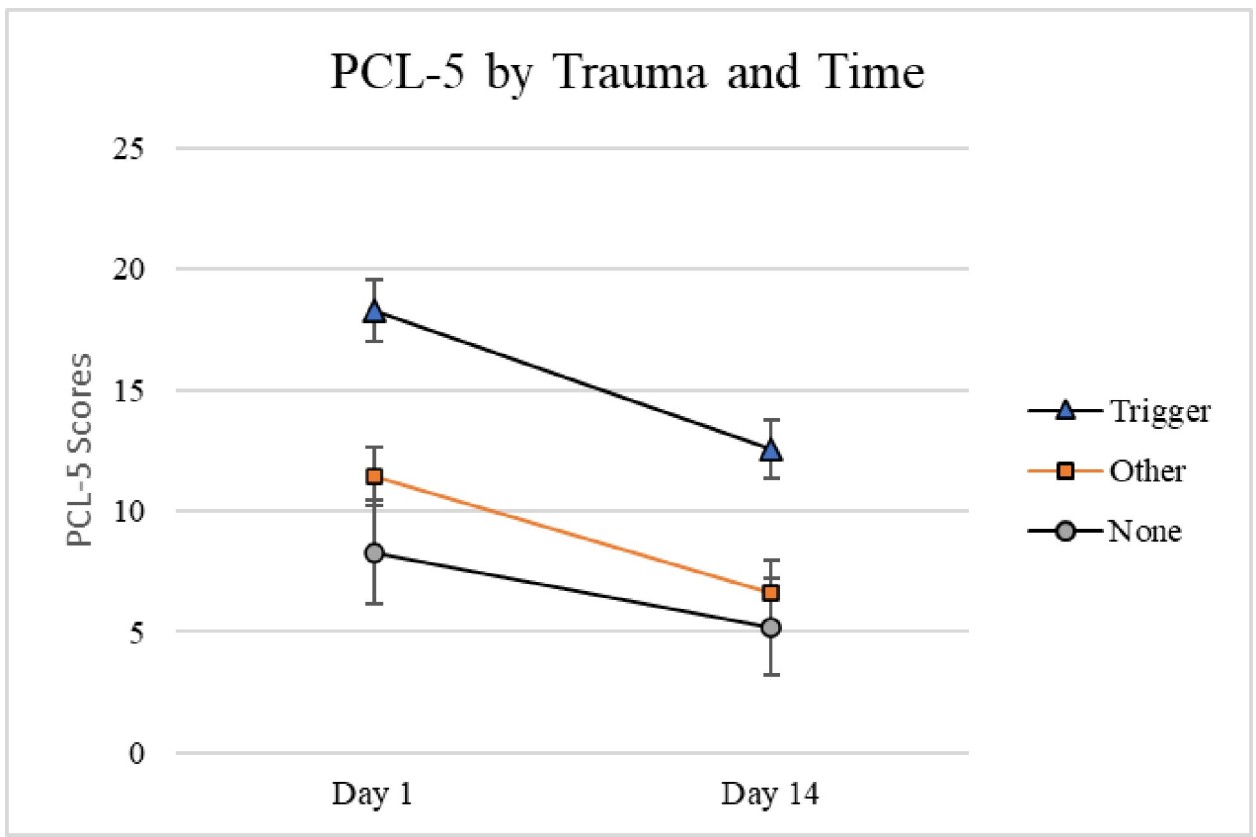

**Fig 3. PCL scores at Day 1 and Day 14 as a function of trauma.** Average PTSD checklist scores shown with standard error bars as a function of day and trauma type (Trigger, n = 95; Other, n = 107; None, n = 34).

## Discussion

The purpose of this study was to investigate how university students reacted to a passage from literature containing depictions of physical and sexual assault. In particular, the study was designed to investigate how one's trauma history and current symptoms affect an individual's responses to such material and how that might change over time.

In this study, the vast majority of participants were willing to read potentially triggering material. Right before reading the passage, all participants who were assigned to the triggering passage were given the offer, made in writing, to read an alternative passage that did not have any content related to physical or sexual assault. At the conclusion of Day 1, participants were asked to indicate, by checking a box, whether they read the assigned or alternate passage. It was clear to the participants that there was no penalty for making this change nor, given the anonymity of the study, was there any way to know if they made the change.

The majority of student participants decided to read the triggering passage even when they were aware of the content. Ninety-six percent of the students who were assigned the triggering passage decided to read it. These percentages were nearly identical for those who had identified themselves as having experienced a triggering trauma (96.9%) or those with a provisional PTSD diagnosis (97.6%). In the qualitative responses, only one participant specifically said they read the alternative passage because "I was afraid that I would be triggered emotionally by the material." Three indicated that "I preferred not to read the more difficult passage because I expected it to be unpleasant". One endorsed both of the previous responses. The most

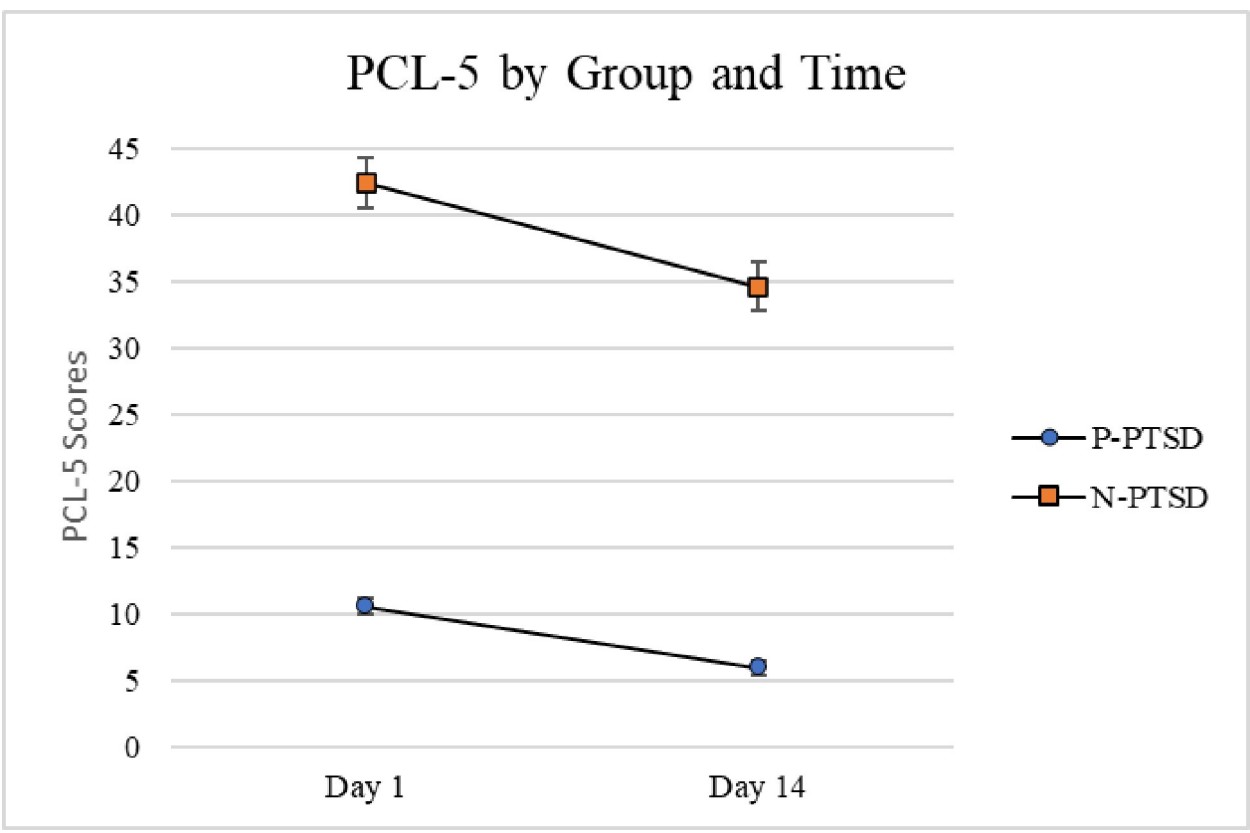

**Fig 4. PCL scores at Day 1 and Day 14 as a function of P-PTSD.** Average PTSD checklist scores shown with standard error bars as a function of day and provisional PTSD (n = 24) and No PTSD (n = 212) groups.

frequently endorsed reason participants chose the alternative passage was it "seemed more interesting". One read the alternative passage because "I could not understand which was the assigned reading", one read the alternative passage because they "didn't want to read something depressing or sad", and two gave no reason for reading the alternative passage.

These data run counter to arguments that suggest that trigger warnings and triggering material will lead students to avoid classes, assignments, or academic responsibilities in numbers that complicate teaching or interfere with instructor choice of assignments. While we could not investigate the question in this study, it is also possible that the de facto trigger warning (via the informed consent procedure) may have increased participation as students felt prepared for the material or were curious about the assigned content. Given the low numbers in this study of students who avoided the material, it would seem that the avoidance of assignments due to possibly triggering material in classrooms would be relatively small particularly due to the pressure some students might feel to complete assignments.

We were surprised to find that there were no differences in avoidance between those with triggering traumas and/or P-PTSD. At a minimum, we expected that individuals with P-PTSD, which include symptoms of behavioral avoidance, would avoid triggering material in a context where there was no penalty for doing so. Perhaps, consistent with research on conformity, they proceeded with the study simply because they were asked to do so. They may have perceived reading the alternative passage as somehow not complying with the request of the experimenters. It is also possible that their general familiarity with the text of a well-known novel may have ameliorated their concern or piqued their interest. It also should be considered

that participants may not have avoided the material because it was written as opposed to material that is audio-visual (i.e., movie) or material presented orally in class (i.e., lecture). A participant would know that a written document can be easily avoided if needed, either by skimming the material or avoiding it altogether by turning to the alternative passage. Some may have started the passage with an intention of not finishing if it became too emotionally intense.

It is probable that avoidance of such material would be even lower in real academic situations. Authorized avoidance of an assignment is typically associated with having to talk to an instructor or a dean, a labor-intensive task that may be emotionally challenging. Avoidance of assigned material without accommodation would typically be associated with a penalty that would impact one's grade, which would further discourage avoidance. For these reasons, it seems reasonable to conclude that avoidance in academic settings is likely to occur less frequently than occurred in this study (~3%) and therefore fairly small in comparison to other reasons students do not go to class or complete assignments.

Therefore, the data suggest that the large majority of students, even those with trigger traumas and P-PTSD, are willing to read triggering material. How students responded to that material on the day they read it however varied as a function of PTSD status but not trauma type. In particular, those with more PTSD symptoms reported higher scores on the RRPQ-R than those without, but having the triggering experience itself was not sufficient to result in higher RRPQ-R scores. These RRPQ scores are largely consistent with the pattern in the SUDS scores. Those with triggering traumas did not necessarily report more distress after reading the triggering passage. In fact, all three groups reported more distress (range 5.1 to 5.6) right after reading the passage then they did two days (range 2.4–2.9) and fourteen days (range 1.9–2.1) later. In addition, the groups did not differ in their reported distress at any time point. All participants, regardless of group status, reported some distress right after reading the passage and then significantly less later. And while this is true of those with triggering traumas, it was also true of the other groups; i.e., the upsetting passage was equally distressing to all of those who read it.

The more robust finding was the significant overall Time effect in SUDS scores. All participants reported being more distressed right after doing the reading than they reported 2 days and 14 days later. (See Fig 1). SUDS scores dropped by the largest amount within the first 48 hours, although the difference in SUDS scores between Days 2 and Days 14 was also significant. Within 48 hours, the SUDS of all three groups were below 3 on an 11-point scale. It is reasonable to surmise that this would be the SUDS among a population of college students at any time point and thus might reflect a return to a value close to "normal". Future designs should incorporate a SUDS assessment before reading the passage to get a sense of the baseline and/or how much distress is associated with anticipatory anxiety. In addition, the slope of the distress within the first 48 hours would also be informative. For those concerned about student health, distress that lasted well after the session, perhaps into the next day, would be more concerning than distress that dropped almost immediately after leaving the experimental session.

While the trauma type had little differential effect on SUDS scores, PCL-5 scores did. Those with PCL scores above 33 reported significantly more distress throughout the study at all three time points. The most parsimonious explanation is that those with PTSD, regardless of their participation in the study, would regularly report distress as a result of the ongoing PTSD symptoms they are experiencing.

However, by using the alternative PCL, and asking participants specifically about any symptoms that resulted from the reading itself, we were able to get some sense of whether they experienced any ongoing intrusive or hyperarousal symptoms that the participants attributed directly to the reading. On average, participants reported a trend (p = .064) toward more

symptoms to the triggering passage as compared to the control passage, an effect that may not have been significant due to the small number of participants assigned to the control condition (n = 32). Those with a trigger trauma, at least compared to those with other traumas, as well as those with provisional PTSD scores reported more symptoms tied to the passage than did those without PTSD or other traumas. This suggests that those with trigger traumas and/or PTSD found themselves thinking about the passage after they had left the session. However, it is important to interpret these findings in light of the magnitude of the effect. For example, those with a trigger trauma reported a mean of 3.54 two days after the reading. This would suggest that they endorsed a few items on the 13-item scale corresponding with "a little bit" or "moderately". A mean of 3.54 would suggest that the majority of items were endorsed "not at all". Given that the total score could be as high as 52, this number represents little ongoing rumination about the passage.

However, when scholars, clinicians, and school personnel use the word "trigger," they are more often referring to the notion that a passage might exacerbate ongoing psychiatric symptoms from an individual's personal trauma that is closely related to the triggering material. While it seems there may have been a small increase in thinking about the passage for those with a triggering trauma, the data also suggest that the passage did not trigger an exacerbation of symptoms related to their own trauma. This is reflected in little change between Day 1 and Day 14 PCL scores.

The primary effect was one of time. All participants reported fewer PCL symptoms at Day 14 relative to Day 1 regardless of provisional PTSD status or type of trauma. Over the 14 days of this study, participants who read the triggering passage did not get worse. Those with provisional PTSD scores and assaultive traumas did report more PTSD symptoms at Day 14 than those in other groups, but this is likely a function of their ongoing PTSD and not related to the passage they read.

In fact, PCL scores went down significantly for all groups from Day 1 to Day 14. While one possibility is that the participants decreased in their symptoms as a function of their study participation, this is not the most likely explanation and the effect is more likely due to procedural issues. The first PCL was given in the lab, in person, on paper, and under circumstances in which they were aware, via informed consent, that they would be reading a passage from literature that would contain assault. In comparison, the Day 14 assessment was taken online at the convenience of the participant. Given that there were no significant interactions with time and that there was a drop that occurred across all groups, this suggests that the difference may be due to the differential administration format of the instrument across two time points. In future designs, the Day 1 PCL could be taken on a computer in the lab and thus be similar in format to the PCL given at Day 14. It also possible that Day 1 assessments were elevated due to anticipatory anxiety associated with knowing they were to read a triggering passage, fill out a trauma history, as well as complete a psychological instrument that inquires of their psychological well-being.

## Limitations

There were a number of limitations to the study. First, the conclusions regarding students' reactions to triggering material are limited as to how students respond to a written passage that includes some detail and reference to physical and sexual assault. For the most part, the content was not graphic. However, we chose these passages from the "Bluest Eye" because this novel is widely used in high school and college and is generally considered a central part of the American literature canon. In addition, we did not assess the desire to avoid the triggering passage beyond whether they choose to switch to the neutral passage. A measure of some desire to

avoid the passage, even if they continued to read it, would have been additionally valuable as would have any previous familiarity with the passage. For example, previous exposure to the passage may have moderated or exacerbated a participant's response. Finally, for the reading of the passage, students came into the lab and spent 30–40 minutes reading the passage in hardcopy and were observed during the process. However, the study did not reliably measure comprehension of the passage and so it is not clear whether all students read the passage completely or skipped sections they found upsetting.

Those students assigned to the control reading were relatively small in number and this may have limited direct comparisons in students' reactions between the control passage and the triggering passage. Given that the main goal of the study was to investigate how individuals with trigger traumas responded to a triggering passage, this discrepancy in assignment was purposeful. Because the base rate of participants with a triggering trauma is fairly low, in order to get a sufficient number of participants in the triggering condition, we had to over-assign participants to the triggering passage. However, this left the power for our control passage versus trigger passage comparison relatively weak and this may have played out in our finding only a statistical trend in the Alt.PCL between the trigger and control passage.

We do not know how students would respond to material that is more graphic, more explicit or presented in a different medium. Films, for example, might produce a different response in students, particularly if they are shown in class where a student might feel that avoidance is difficult or might result in embarrassment. Film and media scholars often argue that film can be more emotionally potent than is the written word. Allowing students to watch films on their own may serve as an easy alternative to circumstances in which a student might feel trapped.

Despite the multi-site characteristics of the study, there were elements of the sample that limit the conclusions we can draw. Even with a fairly large sample, we did not have adequate power to test certain interactions. We had hoped to be able to test a PTSD x Trauma Type interaction to investigate both main effects and interactions. However, some of the cells (such as other trauma/PTSD group and the no trauma/PTSD group) had under 10 participants in each. In this sense, triggering trauma and PTSD were confounded in a manner that made it difficult to assess their separate effects. Assaults typically produce higher PTSD symptoms than do other types of traumas. We do not how the findings would be different, for example, if the triggering trauma passage had been about a significant motor vehicle accident.

In addition, it is difficult to know how our findings may be been impacted by selection bias and how our screening process may have excluded data from participants who may have indeed been "triggered" by the material. Embedded among those who were screened out for drastic changes from Time 1 to Time 2 could have possibly been someone who experienced far more symptoms as a function of their participation. While this is possible, it seems unlikely. Most participants who were screened out had high scores on Day 1 and lower scores on Day 14. In addition, none of the investigators at any of the four sites reported any adverse or unanticipated events. None of the 355 participants who started the study contacted the investigators or their staff about a particularly problematic emotional response to the material. In addition, participants in the study had completed an informed consent in advance and were aware that they could stop at any time without penalty or they could complete the study by reading a control passage. This scenario may feel differently from one that may occur in a classroom where a student may feel compelled to complete the reading or does not feel as if they have a viable option. The sense of control the participants may have had during this study may have minimized distress to the passage relative to a classroom situation. A student who feels as if they have to complete an assignment in order to do well in a class, and is understandably reluctant

to discuss this with an instructor, may feel trapped in a manner that could exacerbate their response to a passage.

The test-retest reliability of the PCL in this sample was acceptable with a correlation of r = .79. This is on par with correlations of the PCL-5 in other samples [27–29]. However, there was a distinct downward trend in the data from Day 1 to Day 14 that is likely procedural in nature. The differing format (hardcopy vs. online), setting (in lab vs. at home), and context may have all made a difference. The Day 1 PCL-5 was given right after the informed consent and right before a trauma checklist, a PTSD measure, and the reading. None of these factors were present during the Day 14 assessment.

## Implications

Students, including those with relevant traumas and PTSD, do not largely avoid potentially triggering passages. There was little evidence that students will avoid assigned material because of its content. Most students showed a willingness to engage with the material, and were distressed briefly after the reading, but the distress did not differ based on trauma history. Finally, the triggering reading did not seem to increase PTSD symptoms in those who started the study with high PTSD scores, and thus the reading did not seem to trigger an exacerbation of their PTSD. This suggests that these types of readings are likely to have brief effects, at least in cases when warnings are presented.

## Acknowledgments

We are particularly grateful for the research support of Lynn Korsun and Kaitlin Morehead at Bucknell University, Spencer Knowlton, Melissa Margolis, and Rachel Semple at California Polytechnic State University, San Luis Obispo and Rebecca Doucet, Anna Kim, and Edith Lopez at Middlebury College.

## Author Contributions

**Conceptualization:** Matthew Kimble, William Flack, Cynthia Meyersburg.

**Data curation:** Matthew Kimble, William Flack, Jennifer Koide, Kelly Bennion, Miranda Brenneman, Cynthia Meyersburg.

**Formal analysis:** Matthew Kimble.

**Funding acquisition:** Matthew Kimble.

**Investigation:** Matthew Kimble, William Flack, Jennifer Koide, Kelly Bennion, Miranda Brenneman, Cynthia Meyersburg.

**Methodology:** Matthew Kimble, William Flack, Jennifer Koide, Cynthia Meyersburg.

**Project administration:** Matthew Kimble, William Flack, Jennifer Koide, Kelly Bennion, Cynthia Meyersburg.

**Resources:** Matthew Kimble.

**Software:** Matthew Kimble.

**Supervision:** Matthew Kimble, William Flack, Jennifer Koide, Miranda Brenneman, Cynthia Meyersburg.

**Writing – original draft:** Matthew Kimble, William Flack.

**Writing – review & editing:** Matthew Kimble, William Flack, Jennifer Koide, Kelly Bennion, Cynthia Meyersburg.

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
