## [Editor Report · Decision Letter 0]

16 Dec 2020

PONE-D-20-37818

Student Reactions to Traumatic Material in Literature: Implications for Trigger Warnings

PLOS ONE

Dear Dr. Kimble,

Thank you for submitting your manuscript to PLOS ONE. After careful consideration, we feel that it has merit but does not fully meet PLOS ONE’s publication criteria as it currently stands. Therefore, we invite you to submit a revised version of the manuscript that addresses the points raised during the review process.

ACADEMIC EDITOR: My comments are attached. Please address all the points raised below.

We look forward to receiving your revised manuscript.

Kind regards,

Kenta Matsumura

Academic Editor

PLOS ONE

Additional Editor Comments:

The authors have attended to the concerns expressed in the previous round of review comments in the revised manuscript. The revised manuscript is much improved—thank you. However, as a new academic editor, I believe that the following issues should also be addressed before this manuscript is considered for publication.

Of particular interest is what would have happened if the “accepting trigger warnings” group had been exposed to the physical and sexual assault passage against their will. There was no experimental condition to test this scenario, so the analyzed data were derived from participants who decided to read the physical and sexual assault passages of their own free will, despite the trigger warning. This point should be addressed in the discussion. This is a potential issue of selection bias, which often occurs in the experimental settings. In addition, please clearly distinguish in the abstract which results were derived from all participants (e.g., the rate of ≥96%) and which were derived from those who decided to read triggering passages of their own free will, despite the trigger warning. Please do not combine these two groups.

“Fourteen participants elected to read the alternative passage.”(P.16): How did the authors treat this group in the analysis? It seems like they were initially assigned to the “trigger” group by the experimenter, but were actually exposed to “control” group stimuli of their own free will. Please specify the treatment of this group (e.g., excluding them from the Fig 1-4 analysis; including them in the “trigger” group). In addition, please consider re-analyzing or conducting additional analyses to assign them to a third category, such as “non-triggering alternative,” rather than including them in the “trigger” or “control” groups, if possible.

Please add the effect size to every result, such as t(23)=4.03, p<.01, Cohen’s d=0.78; F(1,20)=0.15, n.s., partial eta squared=0.02.

The authors used a mixed-design ANOVA. Please conduct Mauchly's sphericity test and apply the G-G or H-F correction where appropriate.

Each figure needs to be independent from the main text and have a title and caption. Abbreviations should be defined. For example, does the error bar represent SD or SEM? Further, please add the number of participants assigned to each group (e.g., Trigger (n=…), Other (n=…)).

Please add the number of participants wherever possible. For example, “There was no significant difference in PCL scores on Day 1 between those who dropped out and those who did not…”(P. 16) should be “… who dropped out (n=…) and those who did not (n=…) …”

Please consider using Tukey’s HSD test or Ryan’s method instead of using the LSD post hoc test. The authors should be aware of the risk of Type I error in multiple comparison.

The authors state, “The majority of participants (n=319) were randomly assigned to the “Trigger Passage” as opposed to…”(P. 13), but “Of the 316 individuals assigned to read the triggering passage,…”(P. 16). Why do these two numbers differ?

Which statistical software did the authors use?
---

## [Author Response · Author response to Decision Letter 0]

30 Jan 2021

Dr. Matsumura,

Thank you for your recent comments and the ongoing work on our manuscript. Please find my response to your most recent editorial requests. 

Comment: “Of particular interest is what would have happened if the ‘accepting trigger warnings’ group had been exposed to the physical and sexual assault passage against their will. This point should be addressed in the discussion.

Response: This is a fair point and has been addressed in the discussion on page 30. There is no way to really know how participants would have responded if they had been forced to read the passage nor is there a way to ethically experimentally test that condition. It is a limitation of the study as students in a classroom may feel compelled to read the passage in a way that participants in this study did not. This could result in those students feeling trapped and could possibly exacerbate their responses. 

Comment: In addition, please clearly distinguish in the abstract which results were derived from all participants (e.g., the rate of ≥96%) and which were derived from those who decided to read triggering passages of their own free will, despite the trigger warning.

Response: We have made that clarification in the Abstract and in other places in the manuscript as well in order to reduce confusion.

Comment: “Fourteen participants elected to read the alternative passage.”(P.16): How did the authors treat this group in the analysis? It seems like they were initially assigned to the “trigger” group by the experimenter, but were actually exposed to “control” group stimuli of their own free will. Please specify the treatment of this group (e.g., excluding them from the Fig 1-4 analysis; including them in the “trigger” group). In addition, please consider re-analyzing or conducting additional analyses to assign them to a third category, such as “non-triggering alternative,” rather than including them in the “trigger” or “control” groups, if possible.

Response: In the end, these fourteen participants were not included in either group because they were assigned to one group but chose to be in another. Their reasons for choosing the alternate reading varied greatly from “Could not understand which one was the assigned reading” to “Found the alternative reading to be more interesting”. Because of this variability, we believe they do not make a cohesive group and therefore it is preferable to assigning them to a third group which would be small and very heterogeneous. 

Comment: Please add the effect sizes to every result

Response: We have done so, either eta squared or Cohen’s d as appropriate

Comment: Please conduct Mauchly’s test…where appropriate.

Response: We have made these changes as well.

Comments: Each figure needs to be independent from the main text and have a title and caption. Abbreviations should be defined. For example, does the error bar represent SD or SEM? Further, please add the number of participants assigned to each group (e.g., Trigger (n=…), Other (n=…)).

Response: We have included the necessary changes to the figures.

Comment: Please add the number of participants wherever possible

Response: Done, and this is particularly valuable in cases in which we were only looking at responses to the triggering passage, thus significantly reducing the n of that analysis compared to the larger sample.

Comment: Please consider using Tukey’s HSD test or Ryan’s method instead of using the LSD post hoc test. The authors should be aware of the risk of Type I error in multiple comparison.

Response: We used Tukey’s as recommended and this more conservative test did change one result for the RRPQ-R that required a change in the manuscript in terms of the interpretation and the results. This change is now reflected in the Abstract, Results, and Discussion (on page 23). We are grateful for the suggestion as they LSD test had resulted in a finding that was actually quite challenging to interpret. Now the RRPQ-R findings line up cleanly with the SUDS findings in a way that one would expect.

Comment: The authors state, “The majority of participants (n=319) were randomly assigned to the “Trigger Passage” as opposed to…”(P. 13), but “Of the 316 individuals assigned to read the triggering passage,…”(P. 16). Why do these two numbers differ?

Response: In order to generate the Tukey’s, the effect sizes, the Greenhouse-Geisser corrections for the paper, we have re done all the analysis for this revision. These numbers are consistently 319. We can’t account for the earlier discrepancy. 

Comment: What statistical software did the authors use?

Response: We used SPSS 27 for the analysis and have included this detail in our manuscript. 

Sincerely

Matthew Kimble, Ph.D.

---

## [Editor Report · Decision Letter 1]

10 Feb 2021

Student Reactions to Traumatic Material in Literature: Implications for Trigger Warnings

PONE-D-20-37818R1

Dear Dr. Kimble,

We’re pleased to inform you that your manuscript has been judged scientifically suitable for publication and will be formally accepted for publication once it meets all outstanding technical requirements.

Kind regards,

Kenta Matsumura

Academic Editor

PLOS ONE

Additional Editor Comments:

The authors treated all the comments adequately, thank you.
---

## [Editor Report · Acceptance letter]

26 Feb 2021

PONE-D-20-37818R1 

Student Reactions to Traumatic Material in Literature:Implications for Trigger Warnings 

Dear Dr. Kimble:

I'm pleased to inform you that your manuscript has been deemed suitable for publication in PLOS ONE. Congratulations! Your manuscript is now with our production department. 

Kind regards, 

on behalf of

Dr. Kenta Matsumura 

Academic Editor

PLOS ONE